# A Clinical Approach for the Use of VIP Axis in Inflammatory and Autoimmune Diseases

**DOI:** 10.3390/ijms21010065

**Published:** 2019-12-20

**Authors:** Carmen Martínez, Yasmina Juarranz, Irene Gutiérrez-Cañas, Mar Carrión, Selene Pérez-García, Raúl Villanueva-Romero, David Castro, Amalia Lamana, Mario Mellado, Isidoro González-Álvaro, Rosa P. Gomariz

**Affiliations:** 1Departamento de Biología Celular, Facultad de Biología y Facultad de Medicina, Universidad Complutense de Madrid, 28040 Madrid, Spain; yashina@ucm.es (Y.J.); irgutier@ucm.es (I.G.-C.); macarrio@ucm.es (M.C.); selene@ucm.es (S.P.-G.); ravillan@ucm.es (R.V.-R.); dcastr01@ucm.es (D.C.); amaliala@ucm.es (A.L.); gomariz@ucm.es (R.P.G.); 2Departamento de Inmunología y Oncología, Centro Nacional de Biotecnología (CNB)/CSIC, 28049 Madrid, Spain; mmellado@cnb.csic.es; 3Servicio de Reumatología, Instituto de Investigación Médica, Hospital Universitario La Princesa, 28006 Madrid, Spain; isidoro.ga@ser.es

**Keywords:** vasoactive intestinal peptide, VPAC1 receptor, VPAC2 receptor, rheumatic diseases, inflammatory bowel disease, central nervous system diseases, type 1 diabetes, Sjögren’s syndrome, biomarkers

## Abstract

The neuroendocrine and immune systems are coordinated to maintain the homeostasis of the organism, generating bidirectional communication through shared mediators and receptors. Vasoactive intestinal peptide (VIP) is the paradigm of an endogenous neuropeptide produced by neurons and endocrine and immune cells, involved in the control of both innate and adaptive immune responses. Exogenous administration of VIP exerts therapeutic effects in models of autoimmune/inflammatory diseases mediated by G-protein-coupled receptors (VPAC1 and VPAC2). Currently, there are no curative therapies for inflammatory and autoimmune diseases, and patients present complex diagnostic, therapeutic, and prognostic problems in daily clinical practice due to their heterogeneous nature. This review focuses on the biology of VIP and VIP receptor signaling, as well as its protective effects as an immunomodulatory factor. Recent progress in improving the stability, selectivity, and effectiveness of VIP/receptors analogues and new routes of administration are highlighted, as well as important advances in their use as biomarkers, contributing to their potential application in precision medicine. On the 50th anniversary of VIP’s discovery, this review presents a spectrum of potential clinical benefits applied to inflammatory and autoimmune diseases.

## 1. Introduction

The nervous, endocrine, and immune systems are coordinated to maintain the homeostasis of the organism, generating bidirectional communication through shared mediators and receptors [1,2].

Vasoactive intestinal peptide (VIP) is the paradigm of an endogenous neuropeptide produced during autoimmune responses and processes of systemic and local inflammation. It acts as an immunomodulatory agent to restore homeostasis of the immune system [3]. Synthesized by neurons and endocrine and immune cells, VIP is involved in the control of both innate and adaptive immune responses [4,5,6]. Exogenous administration of VIP exerts therapeutic effects in models of autoimmune/inflammatory diseases mediated by two G-protein-coupled receptors (VPAC1, VPAC2) [7,8,9,10,11,12].

Inflammatory and autoimmune diseases include a clinically heterogeneous group of chronic diseases sharing inflammatory mechanisms, as well as a deregulation of the immune system [13,14]. These diseases can affect any organ or system and are often multiorganic. Among these pathologies, we find rheumatic diseases such as rheumatoid arthritis (RA), inflammatory bowel diseases (IBD), and multiple sclerosis (MS).

According to the Autoimmune Diseases Coordinating Committee of the National Institutes of Health in the United States, the prevalence of autoimmune pathologies is estimated at up to 8% of the population. These pathologies are characterized by a complex etiology combining different genetic, epigenetic, and environmental factors, such as tobacco use or history of infections, which result in the alteration of the regulation of the immune system [14,15,16,17]. These diseases lead to substantial levels of morbidity, a significant reduction in the quality of life, and premature death [18,19,20,21].

Currently, there are no curative therapies for inflammatory and autoimmune diseases, and patients present complex diagnostic, therapeutic, and prognostic problems in daily clinical practice due to their heterogeneous nature. Many of these challenges could be alleviated with appropriate biomarkers, allowing a more efficient use of current therapies, as well as the development of precision medicine.

This review focuses on the biology of VIP and VIP receptor signaling, as well as its protective effects as an immunomodulatory factor. Here, we consider their role in the pathogenesis of autoimmune diseases and inflammatory disorders and address the potential clinical application of the VIP/receptor axis.

## 2. Biological Characteristics of VIP

### 2.1. VIP Discovery, Cellular Location, and Structure

Sami Said described, for the first time in 1969, the existence of a peptide vasoactive agent with systemic vasodilator capacity present in the lungs of mammals. In collaboration with Viktor Mutt, Said purified this peptide from pig lungs, but only partially. Challenges in isolating it from the lungs led them to examine the intestine, since both tissues have a common embryonic origin. Thus, using porcine duodenal tissue, they isolated this vasodilator peptide and presented it to the scientific community, calling it the vasoactive intestinal peptide [22].

A few years later, the presence of this peptide was demonstrated in different areas of the central and peripheral nervous system, such as the bodies, axons, and neuronal dendrites [23], as well as in presynaptic endings [24], resulting in the categorization of the VIP as a neuropeptide with neuromodulatory and neurotransmitter functions. This role was confirmed with the characterization of VIP receptors in numerous areas of the central nervous system (CNS) [25]. 

In the immune system, the first information dates back to 1985, when Felten et al. described VIP-like immunoreactivity in the thymus nerve endings [26]. Since then, VIP-ergic innervation in the spleen, lymph nodes, and mucosal-associated immune system has been demonstrated [27]. It is also important to note that sympathetic nervous system fibers innervate the joints, which explains the role of VIP in rheumatic diseases. 

Regarding the cellular source involved in VIP production, the first evidence was reported for cells of myeloid lineage. Expression in mast cells was demonstrated by radioimmunoassay and immunohistochemistry in the rat peritoneum, intestine, and lung [28]. In 1980, O’Dorisio described the presence of VIP in human peripheral blood polymorphonuclear cells, especially neutrophils, but not in mononuclear cells [29]. VIP expression has also been described in human eosinophils [30] and in eosinophils of granulomatous lesions induced by infection with *Schistosomiasis mansoni* [31]. We reported that neither M1 nor M2 human macrophages express transcripts of VIP [32]. Concerning cells of a lymphoid lineage, in the 1990s, our team reported, for the first time, the synthesis and secretion of VIP in murine T and B lymphocytes [33,34,35]. Since then, information on the important role VIP plays in inflammation and autoimmunity continues to accumulate. Today, VIP is an important player in the circuit formed by the nervous, endocrine, and immune systems. It is also present in rheumatic diseases [36] and is one of the most studied peptides in terms of a physiological role in health and disease, especially in the immune system.

The origin of VIP in the microenvironment of the different pathologies in which its effect has been studied is the nerve endings and cells. In this sense, nerve fibers of the sympathetic nervous system in the joints have been reported in rheumatic diseases. Moreover, a decrease in the number of these nerve endings has been described in osteoarthritis (OA) and RA [37]. Regarding cellular origin, synovial fibroblasts (SF) from OA and RA patients have been found to express and release VIP [38].

VIP belongs to a broad family of neuropeptides and hormones, related both structurally and at sequence level, called the secretin/VIP family. In addition to VIP and secretin, this family includes the adenylate cyclase activating peptide pituitary (PACAP) 27 and PACAP38, helodermin, histidine-methionine peptide (PHM, in humans) or histidine-isoleucine peptide (PHI, in other mammals), the releasing factor of growth hormone (GHFR), glucagon and its related peptides GLP1 and GLP2, and the gastric inhibitor peptide (GIP) [39]. The structural homology observed among the different members of this family is very high, with the following characteristics being common [40]: (I) precursor peptide formed by a signal peptide, from 1 to 3 bioactive peptides and N- and C-terminal peptides; (II) length of the mature peptide comprised of between 25 and 50 aa residues; (III) synthesis and release by nerve, immune, and/or endocrine cells; (IV) patent tendency for the formation of α-helix structures; and (V) presence of a structural motif called N-Cap in the amino terminal region. The helical structure seems to be a key element in the interaction with receptors and signaling and is considered an interesting therapeutic target [41]. These peptides show strong homology in their amino acid sequences on an evolutionary scale, suggesting a common origin from an ancestral gene [42]. VIP is a 3.326 Da molecular weight peptide, with a basic nature and amphipathic character. Its primary structure consists of a single chain with 28 aa whose sequence has been highly conserved throughout evolution [43]. Although the presence of all of these aa is necessary for VIP to perform its biological functions, it has been proven that certain residues are crucial for this performance (His^1^, Val^5^, Arg^14^, Lis^15^, Lis^21^, Leu^23^, and Ile^26^). The secondary structure has a random coil in the N-terminal region and an α-helix structure in the C-terminal region. This structure is similar to that of other family members, especially that of PACAP27, with whom it shares 68% sequence homology [44].

### 2.2. General Biological Functions

The expression of VIP in the nervous system results in its release in multiple organs by releasing nerve fibers. Thus, VIP is present in the innervation of the heart, kidney, lung, thyroid gland, and gastrointestinal and urogenital tracts. As we have described previously, central and peripheral lymphoid organs, such as the thymus, spleen, and lymph nodes, are also innervated by VIP sympathetic nerve fibers. Moreover, VIP expression in cells of myeloid and lymphoid origin also contributes to its broad distribution, correlating with its functional pleiotropism. Thus, VIP acts as a neurotransmitter, immunoregulator, vasodilator, and stimulator of hormone secretion or secretagogue [6]. VIP contributes to a wide variety of physiological activities related to development, growth, immune response, circadian rhythms, endocrine control, and functions of the digestive, respiratory, cardiovascular, and reproductive systems [39]. Some of VIP’s multiple biological activities include increased cardiac output, bronchodilation, smooth muscle relaxation, regulation of secretion processes, and motility in the gastrointestinal tract. In addition, as a secretagogue, VIP promotes the release of prolactin, luteinizing hormone, and growth hormone by the pituitary gland and regulates the release of insulin and glucagon in the pancreas. This peptide also promotes analgesia, hyperthermia, learning, and behavior; it has neurotrophic effects and regulates bone metabolism and embryonic development [45].

## 3. VIP Receptors, Ligands, and Signaling Pathways

### 3.1. VIP Receptors

VIP and PACAP were discovered in the 1970s and 1980s, respectively, and cloned in the 1980s and 1990s. The existence of several “VIP receptors” was inferred by pharmacological studies, such as cyclic AMP (cAMP) or radioligand binding assays, long before the actual receptor cloning. In this way, “VIP receptors” were described in normal and tumor cells and tissues [40]. These receptors showed pharmacological differences and were not correctly identified until the description of several ligands, agonists, and antagonists, and the cloning of the receptors. 

VIP receptors belong to group B of G-protein-coupled receptors (GPCRs), which include seven transmembrane receptors that represent the most extensive family of signaling proteins. Ligands for class B GPCRs are peptides that bind to the large N-terminal part of the GPCR [46]. There are three receptors recognized by VIP: VPAC1, VPAC2, and PAC1 receptors. VIP binds to VPAC receptors with equal affinity and with much less affinity to PAC1, which is the PACAP-preferred receptor. The structures of several class B receptors have been determined, as well as crystal structures of peptide-bound receptors, which help to understand how the peptides can bind to their receptors and how these receptors undergo conformational changes to allow downstream signaling [47].

#### 3.1.1. VPAC1 Receptor 

The rat VPAC1 receptor was cloned in 1991 from a cDNA library, and the human VPAC1 was cloned in 1993 from the HT-29 cell line [48,49]. Only one variant of this receptor has been described thus far, which is expressed in several normal and malignant cells. It has a deletion that results in a receptor with five transmembrane domains lacking the G-protein binding domain. Even so, it can activate protein tyrosine kinase activity, but in a different way than the seven transmembrane domain receptor [50].

The affinity of several peptides for this receptor is as follows: VIP = PACAP > GRF > secretin [40]. In the last decade, advances in the study of molecule structures have allowed the dissection of the physical sites of interaction between VPAC1 and VIP, observing that the side chains of several residues in the VIP sequence are in contact with several others in the receptor sequence, although the whole interaction between the two molecules has yet to be elucidated. Nevertheless, all the models available are in concordance with the mechanism proposed for the ligand–receptor interaction for this family of receptors, the “two domain” model, in which part of the peptide remains inside the N-terminal ectodomain of the receptor, while the N-terminus of the peptide is able to interact with the transmembrane region of the receptor [39].

#### 3.1.2. VPAC2 Receptor 

This receptor was also first cloned from rats, with the human and mouse receptors cloned shortly thereafter [51,52,53]. The first variant of this receptor described in mice tissues showed a deletion in exon 12, which corresponds to the carboxyl-terminal end of the seventh transmembrane domain. This variant lacks its normal function of increasing cAMP [54]. The second variant found in a human malignant T-cell line presented a deletion in exon 11 and had lower affinity for VIP [55]. The most recently described was the same variant as that described for the VPAC1 receptor [50]. The order of affinity for human VPAC2 expressed in different cell lines is VIP = PACAP = helodermin > secretin [40].

### 3.2. Ligands

During the last four decades, many ligands have been developed, both agonists and antagonists for VIP receptors. Most of them were created by modifying endogenous peptides and displayed different affinities and selectivities, with the first descriptions unable to differentiate between the two receptors [56,57,58]. Selective agonists for VPAC1 receptor have been generated, such as [K^15^, R^16^, L^27^]VIP(1-7)/GRF(8-27) [59], [Ala^11,22,28^]VIP [60], [L^22^]VIP [61], [R_16_]PACAP(1-23) [62], and LBT-3393 [63]. A selective antagonist is also available: PG97-269 [64]. Regarding VPAC2, cyclic peptides have been demonstrated to be selective agonists, such as Ro25-1553 [65], Ro25-1392 [66], and some other peptides, such as BAY 55-9837 [67] Hexanoyl [A^19^,K^27,28^]VIP, rRBAYL [68], and LBT-3627 [63]. Only two VPAC2 selective antagonists have been described thus far: PG99-465 [69] and VIpep-3 [70]. Some of these approaches aim to identify more metabolically stable peptides [63] in order to ameliorate their in vivo administration. Recently, one in silico study predicted possible structures defining affinities for these receptors, constructing classifiers to predict the bioactivities of novel VPAC ligands and highlighting the importance of electrostatic properties in the interaction of VIP derivatives with the receptors [71]. Moreover, there are some other approaches providing tools for the study of these receptors or for their use as therapeutic tools, such as nanoparticles that enhance the half-life of one VPAC2 selective agonist [72] and specific nanobodies for the VPAC1 receptor that bind at a different site than VIP, thus without interfering with the coupling of the peptide [73]. 

### 3.3. Signaling Pathways

The main signaling pathway of VPAC receptors is their coupling to G-proteins, which are heterotrimeric proteins composed of three subunits: α, β, and γ. When stimulated, the α subunit binds to GTP and dissociates from the βγ dimer. Activated Gα moves through the membrane to its effector, the enzyme adenylate cyclase (AC), which in turn catalyzes cAMP synthesis [40]. This second messenger classically activates protein kinase A (PKA), which phosphorylates and can activate or inactivate different signaling pathways, depending on the cell type. For instance, the typical transcription factor activated by cAMP through PKA is cAMP-response element binding (CREB). PKA can also activate mitogen-activated protein kinases (MAPK) from several subfamilies. Moreover, cAMP in a PKA-independent way can activate exchange proteins directly activated by cAMP (EPAC), which is a G-protein exchange factor (GEF) for Rap small G-protein [74]. The βγ dimer interacts with several proteins, such as Ras, which in turn activates extracellular regulated kinases (ERK). Subsequently, ERK interacts with and activates phosphoinositide 3-kinase (PI3K) [75,76]. 

VPAC receptors can also mediate the increase in Ca^2+^ through the activation of either Gi/o or Gq proteins. Although this pathway has shown lower potencies, relative to cAMP, the rise in calcium is of physiological relevance [77]. Less frequently, they can also activate phospholipase C (PLC) through the activation of Gi/o by a mechanism that likely involves the βγ dimer and that subsequently increases the production of inositol phosphate (IP_3_). This rise in IP_3_ increases the [Ca^2+^], which, together with diacylglycerol (DAG), activates PKC, which can also phosphorylate several other kinases [78]. Furthermore, PLD can be activated in a pathway involving the small G protein ARF. This phospholipase hydrolyzes phosphatidylcholine (PC), generating the signaling molecule phosphatidic acid (PA), which functions pleiotropically in several signaling pathways [79].

Receptor activity modifying proteins (RAMPs) are single pass transmembrane proteins that do not bind any known ligand and need to be coupled with any receptor to arrive at the cellular surface. VPAC1 and VPAC2 can bind the three known RAMPs, not modifying their affinity for ligands. Both receptors enhance the cell-surface expression of the three RAMPs, and co-expression of VPAC1 and RAMP2 enhances the response of IP_3_ without modifying cAMP signaling [80]. Furthermore, VPAC2 co-expression with RAMP1 increases the basal cAMP and is diminished when VPAC2 is co-expressed with RAMP3. Moreover, co-expression with RAMP1 and 2 enhances coupling to Gi/o/t/z, but does not modify the binding to Gs [81].

Regarding inflammatory pathways, three of the most important transcription factors activated in inflammatory processes are AP-1 (activator protein 1), NFκB (nuclear factor κB), and IRF (interferon regulatory factor). VIP is able to inhibit AP-1 and IRF activation through a PKA-dependent mechanism and can also prevent NFκB translocation to the nucleus impeding IKK activation through a cAMP independent mechanism [82,83,84,85].

The βγ dimer has been shown to bind GPCR kinases (GRKs), recruiting them to the membrane. When GPCRs are phosphorylated by GRKs, they bind arrestins, allowing receptor desensitization and internalization and/or signaling from the inside, mainly through activation of different MAPKs [46]. In particular, VPAC1 and VPAC2 exhibit augmented desensitization when co-transfected with GRKs 2, 3, 5, and 6 [86]. Nonetheless, this is not the only way in which GPCRs have effects within the cell. Recently, the presence of intracellular GPCRs has been described, which arrive there through many different processes [87]. VPAC1 has been found to be expressed in the nucleus of several tumor cells, such as human breast cancer [88], human renal carcinoma [89], and human glioblastoma, where there is also a weak expression of nuclear VPAC2 in one of the cell lines studied [90]. More recently, the presence of VPAC1 has been observed to be located on the surface and nuclear membrane of T helper (Th) cells, whereas its expression is limited to the nucleus when these cells are activated [91]. There is no evidence regarding how VPAC2 arrives at the nucleus, and all the options described elsewhere are possible [87]. However, there is work showing that VPAC1 has a nuclear localization signal sequence in the C-terminal domain of the receptor, and in this way is able to exhibit nuclear expression [90]. It has been described that Cys37 in the VPAC1 receptor is essential for the translocation of the receptor to the nucleus and that it must be palmitoylated to be functional [92].

## 4. A Very Important Peptide in Inflammation and Autoimmunity 

### 4.1. Targeting Balance of Inflammatory Factors

Inflammation is a complex homeostatic process mediated by factors of plasma and cellular origin whereby the effects of harmful stimuli are controlled in the tissues. When the inflammation persists over time, beyond what is necessary, and stops responding to the reparative process, it becomes destructive and chronic.

In chronic inflammation, there is a massive infiltration of cells involved in innate (monocytes–macrophages and dendritic cells) and adaptive immunity (TCD4^+^ and B cells). A complex network of pro-inflammatory cytokines is established, and the cytokines are secreted primarily by activated macrophages and CD4^+^ T cells at the site of inflammation.

The macrophage is the main producer of cytokines, and, when activated by different danger signals, it releases several pro-inflammatory products, such as interleukin (IL)-1, tumor necrosis factor (TNF-α), IL-6 and IL-l2, and nitric oxide (NO), followed later by the secretion of anti-inflammatory cytokines such as IL-10 [93]. At the site of inflammation, numerous chemokines are also secreted, exacerbating the inflammatory process by the attraction of more leukocytes. Despite its beneficial effects in the defense of the organism, the sustained production of pro-inflammatory factors can lead to pathological conditions such as septic shock, respiratory distress syndrome, and autoimmune disease [94,95,96].

Numerous studies, both in animal and human models, show that VIP plays a key role in maintaining homeostasis, by controlling the balance of pro- and anti-inflammatory cytokines by inhibiting the production of pro-inflammatory cytokines and chemokines such as TNF-α, IL-6, IL -12 CXCL8, and CCL2, as well as NO, and stimulating the expression of anti-inflammatory cytokines such as IL-10 [4,5,6].

In activated macrophages, VIP inhibits the production of TNF-α, IL-12, and NO primarily through VPAC1, expressed constitutively, and, to a lesser degree, through inducible VPAC2. VIP’s binding to VPAC1 induces both a cAMP-dependent and a cAMP-independent pathway that regulates cytokine production and NO at the transcriptional level. VIP inhibits the expression of TNF-α, IL-12, and inducible nitric oxide synthase (iNOS) by reducing the binding of the NFκB transcription factor to the promoter and increasing IL-10 by increasing the binding of the CREB factor [97]. Thus, molecular mechanisms and transcription factors involved in the VIP signaling during inflammatory responses include inhibition of interferon (IFN)-γ-induced Jak1/Jak2 phosphorylation and STAT1 activation, inhibition of different MAPK cascades, inhibition of IĸB-kinase, and stimulation of CREB factor [3,97] (Figure 1).

VIP also modulates inflammatory responses through the regulation of different functions of other cells, including mast cells, microglia, dendritic cells, and synovial fibroblasts [98,99,100,101]. Moreover, in terms of adaptive immunity, VIP reduces pro-inflammatory Th1 and Th17 responses, as described below.

The importance of endogenous VIP in the regulation of inflammation and autoimmunity has been confirmed in knockout (KO) mouse models showing altered immune responses. At basal conditions, the immune phenotype of the mice studied so far is relatively mild. The role of VIP is mainly highlighted in challenging inflammatory conditions. Thus, VIP-deficient mice develop lung inflammation [102,103]. However, there are discrepancies about the resistance or susceptibility of VIP-KO mice to endotoxemia. Hamidi et al. described an increased susceptibility to death from endotoxemia, while Abad et al. found that VIP-KO mice exhibit resistance to endotoxic shock and decreased pro-inflammatory responses due, in part, to the presence of an intrinsic defect in the responsiveness of inflammatory cells in the chronic absence of VIP, suggesting that these mice may exhibit a defect in the innate arm of the immune system [104,105]. 

Given the accumulated evidence of VIP anti-inflammatory properties, VIP treatment has been reported to protect against septic shock and various inflammatory and autoimmune diseases, and to act as a survival factor against injury of lung and neuronal cells [45,106,107,108,109,110]. The role of VIP in the inflammatory component of the diseases described in this review is treated in detail in the different pathologies.

### 4.2. Modulating the Expression of TLRs

Toll-like receptors (TLR) are a large family of type I transmembrane proteins belonging to pattern-recognition receptors, which are specialized in the recognition of extracellular and endosomal pathogen-associated molecular patterns, serving as warning signals for the immune system. Likewise, TLRs are able to specifically bind damage-associated molecular patterns, which are associated with tissue damage, cell stress, and cell death [111,112,113]. Therefore, TLRs are defined as essential receptors to trigger innate immune response and, subsequently, for the regulation of the adaptive immune response [114]. The expression of these receptors has been detected not only in immune cells, including macrophages, neutrophils, mast cells, dendritic cells, and T and B lymphocytes [115,116,117], but also in non-immune cells, such as synovial fibroblasts, keratinocytes, pulmonary, and intestinal epithelial cells [118,119,120,121]. In humans, these transmembrane receptors are found both on cell membranes (TLR1, 2, 4, 5, and 6) and in endosomes (TLR3, 7, 8, and 9) [111,122].

Upon ligand binding to TLRs, complex signal transduction cascades are triggered, requiring different adapter proteins. Myeloid differentiation primary response protein (MyD88) is involved in signaling by all TLRs, with the exception of TLR3 [123,124]. Both MyD88-dependent and -independent pathways lead to activation of NFκB, IRF3/7, and/or AP-1, which ultimately induce the production of inflammatory mediators, and co-stimulatory molecules [111,113,125]. Thus, an inappropriate or deregulated TLR activation, such as a persistent infection or a failure in their ability to discriminate self from non-self molecules can compromise immunological homeostasis [114,126]. In fact, numerous studies have demonstrated the involvement of TLRs in a wide variety of pathological processes, including both acute and chronic infections, as well as in the induction, progression, or exacerbation of many systemic autoimmune and/or inflammatory conditions [113,127,128,129]. In this regard, extensive data accumulated from animal models and in vitro human studies have strongly demonstrated the homeostatic effects of VIP on the deregulated expression and signaling of TLR in a context of inflammatory and/or autoimmune disease [130,131,132,133].

TLR modulation by VIP was described for the first time in the trinitrobenzene sulfonic acid (TNBS)-induced colitis mouse model, which mimics human Crohn’s disease (CD). In that model, VIP reduces the upregulated expression of TLR2 and TLR4, as we describe in Section 5, “Protective Effects of VIP in Inflammatory/Autoimmune Diseases” [134,135]. The inhibitory effect of VIP on TLR2 expression was suggested to be due to its ability to prevent the nuclear translocation of NFκB, which has a binding site in the murine TLR2 gene [85,136]. Moreover, research on primary murine macrophages and the RAW 264.7 cell line showed that VIP exerts its suppressive effects on murine TLR4 expression at the transcriptional level by decreasing the binding of the transcription factor PU.1 via PI3K/Akt1 pathway [137]. In agreement with these findings, VIP was also able to reduce the lipopolysaccharide (LPS)-induced expression of TLR2 and TLR4 in human monocytic THP1 cells and peripheral blood monocytes, as well as to inhibit their differentiation to macrophages [138]. In these cells, VIP inhibited the nuclear translocation of PU.1, which acts as a transcriptional regulator of both TLR2 and TLR4 genes in humans [139,140]. 

The potent immunomodulatory effect of VIP on TLRs has also been reported in the mice cornea after *Pseudomonas aeruginosa* infection. Data from in vitro studies demonstrated that VIP reduced LPS-stimulated expression of TLR1, TLR4, TLR6, TLR8, and TLR9 in macrophages and Langerhans cells [141]. 

The effects of VIP on TLRs were also assessed in SF from OA and RA patients. TLR2, TLR4, and TLR3 expression are described to be higher in RA-SF compared with OA, whereas greater levels of TLR7 have been detected in OA-SF [82,142,143]. In vitro data indicated that VIP treatment decreases both LPS- and TNF-induced expression of TLR4 in RA-SF, whereas it has no effect on the elevated constitutive expression of TLR2 and TLR4 [142,144]. Furthermore, VIP also exerts a negative modulation of TLR4 signaling in these cells by the downregulation of important molecules of both the MyD88-dependent and -independent signaling pathways [83]. On the other hand, no effect of VIP on the expression of other TLRs has been detected in OA and RA-SF. However, VIP exerts an inhibitory activity on nuclear translocation of transcription factors activated by TLR3 and TLR7, with the subsequent reduction of antiviral, pro-inflammatory, and joint destruction mediators upregulated by engagement with these receptors [82,142].

All in all, VIP’s ability to balance TLR expression and signaling may be of physiological relevance in the specific control of innate and adaptive immune responses.

### 4.3. Regulating Th Cells

Th cells are specialist cytokine-producing cells that modulate the adaptive immune response. During inflammation or infection, different Th subsets are activated, playing a fundamental role in the type of response and the degree of amplification. These subsets are classified by their cytokine profile and the expression of specific transcription factors (master regulators) that direct their functional activity [145]. Thus, Th subpopulations are organized into two branches, effector Th cells and regulatory T cells (Treg). Th1, Th2, Th17, Th follicular (Tfh), Th9, and Th22 subsets are found within the branch of effector Th cells. In an immune steady-state, the balance between these subsets underwrites the preservation of immune tolerance. When a microbial or viral infection or tissue damage occurs, this balance changes from a tolerant state to an immunogenic/inflammatory state, until the immunogen is eliminated. Then the homeostatic regulatory mechanisms are recovered, and the system returns to its initial state. Inflammatory and/or autoimmune diseases occur when these mechanisms fail [146]. Some of the Th subpopulations play a key role in these pathologies; for example, Th1 and Th17 are the key effector Th cells in RA and Crohn’s disease, while a loss in the number or function of the Treg has been described. Not only is the level of presence of each of the subsets important in the development of autoimmune diseases, but so is the plasticity observed between them or even their heterogeneity [147,148,149,150]. In this sense, pathogenic Th17 can change its lineage commitment to a Th1 profile, called nonclassical Th1 or ex-Th17. This has been observed in different mouse models of autoimmune diseases or in RA patients [151,152,153,154]. Th plasticity is also observed in Treg, which can shift its linage commitment to Th1 or Th17. In turn, nonpathogenic Th17 can acquire a Treg profile [150]. Therapeutically, it is important to know the involvement of each subpopulation in the different pathologies, as well as their possible plasticity or heterogeneity. 

VIP is a microenvironment mediator involved in the generation of diversity and plasticity of Th subsets in inflammatory or autoimmune diseases. This claim is supported by numerous experimental studies in both animal models and ex vivo samples of patients [5,155,156,157,158]. VIP was able to decrease the cytokine profile and master regulators related to Th1 and Th17 subsets and to increase those of them related to Treg or Th2 in different autoimmunity animal models, such as the collagen-induced (CIA) arthritis mouse model of RA, the TNBS mouse model of Crohn’s disease, the nonobese diabetic (NOD) mouse model of autoimmune diabetes, the experimental autoimmune encephalomyelitis (EAE) mouse model of multiple sclerosis, the experimental model of autoimmune myocarditis, and the pristine-induced lupus model of lupus nephritis [7,11,159,160,161,162,163,164,165]. In addition, this immunomodulatory role of VIP was observed in two inflammatory animal models, including the models of CNS inflammation or atherosclerosis [108,164]. The same effect was observed in mouse Th cells activated in vitro studies or with Th lymphocytes from patients activated ex vivo, mainly in studies with RA patients, treated with exogenous VIP [153,157,166,167]. VIP not only acts on a specific subset in these pathologies, but is also able to balance the different Th subsets, inducing nonpathogenic phenotypes or modify their plasticity. Studies on different transcription factors, cytokines, cytokine receptors, chemokines, and chemokine receptors in the above mentioned mice models, in vitro or ex vivo, showed that VIP counterbalances the ratio of Th1/Th2, Th17/Treg, Th1/Treg, or Th2/Th9, reducing pathogenicity and increasing tolerance [10,165,168]. Th17 cells are a heterogeneous subset with a nonpathogenic or pathogenic profile, depending on the microenvironment. VIP maintains the nonpathogenic profile of human Th17-polarized cells in vitro from naïve Th cells [169]. Indeed, it lowers the pathogenic Th17 profile in activated/expended memory Th cell ex vivo from early RA patients [153,170]. Taking into consideration the plasticity of Th subsets, this neuropeptide decreases the Th17/1 profile, inducing a negative correlation between Th17 and Th1 in ex vivo cultured cells from early RA patients, but also increases the Th17/Treg profile [153,169,170]. The effect of VIP on the plasticity of Th17 cells is in agreement with its effect on heterogeneity, since the nonpathogenic Th17 phenotype is closely related to Th17/Treg plasticity.

In summary, the generation/differentiation, plasticity, and heterogeneity of Th subsets are crucial events during the development of inflammatory/autoimmune diseases. These processes are susceptible to modulation by different mediators present in the microenvironment of Th cells, an example of which is the VIP neuropeptide that induces a less pathogenic and more tolerogenic response in Th cells.

### 4.4. Inducing Tolerogenic Dendritic Cells

Conventional or classical dendritic cells (DCs) are critical for initiating the activation and differentiation of T cells during an inflammatory state, mainly due to their co-stimulatory capacity. They can be classified functionally according to their maturation state in immature or mature DCs [146]. Immature DCs, also called lymphoid organ-resident DCs, are phenotypically immature since they show on their surface low amounts of costimulatory receptors. When they migrate, they initiate a maturation process by strongly expressing these receptors. During the maturation process of DCs, they can differentiate into tolerogenic or immunogenic antigen-presenting cells, each distinguished by specific cytokine production and cell-surface receptors. Immunogenic DCs develop an immunogenic/inflammatory state, whereas mature tolerogenic DCs can induce immune tolerance [146,171]. These latter cells are prompted by either anti-inflammatory signals or signals interfering with the function of immunogenic DCs. Their role is to inhibit effector and autoreactive T cells and trigger Treg development. As a consequence, they play a main role in inducing immune tolerance, resolution of ongoing immune responses, and prevention of autoimmunity [172,173]. 

An increasing body of data indicates that tolerogenic DCs could be promising therapeutic targets in the treatment of autoimmune diseases [172,174]. One of the approaches is to generate ex vivo tolerogenic DCs for DC-based immunotherapy [175,176]. In this sense, VIP-treated DCs retained their tolerogenic ability in vitro and in vivo under different inflammatory situations [45,177,178]. Two strategies have been followed to generate VIP-tolerogenic DCs: VIP treatment during differentiation of DCs derived from bone marrow or monocytes, or using lenti-VIP transduced DCs [6,179]. In either case, the later administration in vivo of these cells produces Ag-specific Treg capable of inducing specific tolerance to naïve recipients. In this way, they cause the attenuation of symptoms of different animal models of autoimmune and/or inflammatory diseases, for example, in CIA arthritis, TNBS-induced colitis, EAE, sepsis, and spontaneous autoimmune peripheral polyneuropathy [177,179,180,181,182]. In addition, in vitro studies with VIP have shown that it affects not only the phenotypic and functional maturation of DCs, but also the migration of these cells [183,184,185]. 

## 5. Protective Effects of VIP in Inflammatory/Autoimmune Diseases

### 5.1. Rheumatic Diseases

#### 5.1.1. VIP in Rheumatoid Arthritis

RA is a systemic inflammatory rheumatic disease of unknown etiology, with a significant autoimmune component, characterized by a persistent synovitis of symmetrical peripheral joints and the presence of auto-antibodies such as rheumatoid factor and anti-citrullinated protein antibodies (ACPA) [186,187,188]. The natural course of RA is generally associated with progressive destruction of articular cartilage and bone, resulting in a severe functional impairment and serious worsening of the patient’s quality of life. However, RA is described as a heterogeneous disease with several subtypes that differ in clinical symptoms, such as age of onset, rate of progression, disease severity, and outcome [187,189,190]. Its complexity is also reflected in the fact that it is considered a multifactorial disease, as genetic background and environmental conditions, including infectious events and dysbiosis in the gut and the lung microbiome, have been indicated as factors involved in triggering the aberrant immune response [186,187].

Although RA pathogenesis is not completely understood, it is widely accepted that local and systemic immune dysregulation, as a result of imbalance in the Th cell subsets, plays an important role in creating a synovial joint microenvironment that favors a hyperactivated phenotype of SF and macrophages. Indeed, both cell types are thought to be central to disease progression by mediating synovial hyperplasia and the release of pro-inflammatory cytokines and tissue damaging enzymes [187,191,192,193,194]. In RA, resident synovial cells and both adaptive and innate immune cells establish a positive feed-forward activation loop mediated by pro-inflammatory cytokines such as TNF-α, IL-1β, IL-6, and IL-12, which perpetuate the disease and ultimately lead to joint destruction [195,196,197,198].

Experimental evidence accumulated over the last two decades has demonstrated beneficial effects of VIP in all stages of RA development through its anti-inflammatory and immunomodulatory abilities [131,158,199] (Figure 2). In addition, numerous studies have shown the direct antimicrobial activity of VIP against a wide range of bacteria [200], as well as its protective effect in polymicrobial sepsis [201]. Interestingly, VIP is able to counteract the effects of LPS from *Porphyromonas gingivalis* in monocytes, which is an oral bacterium related to increased risk of arthritis associated with periodontal disease [202,203]. 

Initial data about the anti-inflammatory properties and therapeutic potential of VIP in the context of RA were obtained in the CIA mouse model [11,199]. Exogenous administration of VIP was able to reduce the incidence and severity of arthritis in mice, inducing a dramatic decrease in cartilage and bone erosion. In that model, VIP has been demonstrated to modulate the subsets of Th lymphocytes by promoting a Th2-type response while expanding CD4^+^ CD25^+^ Treg [11,204]. In line with these findings, other studies in the same arthritis mouse model showed protective effects of VIP on bone destruction by modulating the RANK/RANKL/OPG system through the downregulation of the Th17 response and subsequent increase of the Treg/Th17 ratio [84,159,168]. Moreover, an inhibitory action of VIP on osteoclastogenesis has been described in CIA mice, exerting a direct effect on osteoclast progenitor cells purified from bone marrow, as well as through its modulatory action on stromal and osteoblast cells [84,205]. 

In light of the anti-inflammatory and immunomodulatory effects of VIP in the CIA model detailed above, several studies assessed the role of this neuropeptide in the context of human RA. Accordingly to the murine model, in vitro studies on human synovial fibroblasts, macrophages, peripheral blood lymphocytes, and Th cells from patients with RA confirmed the ability of VIP to regulate components of both innate and adaptive immune responses [158]. 

In brief, VIP has been shown to significantly attenuate the basal and TNF-α-induced production of pro-inflammatory chemokines and IL-6 in both synovial tissue suspensions and SF from RA patients [98]. Interestingly, such anti-inflammatory effects were later fully reproduced in cultured RA-SF by specific VPAC2 agonists, according to the dominant presence of that receptor described in these cells [38]. 

Subsequent studies on RA-SF also proved an inhibitory effect of VIP on the expression and signal transduction of some PRRs, which are linked to the pathogenic activation of these synovial cells [193,197] as previously explained. Moreover, VIP is able to downregulate the enhanced expression of the IL-22 specific receptor, preventing the IL-22 stimulatory effects on proliferation and production of matrix metalloproteinase-1 (MMP-1) and S100A8/A9 alarmins involved in RA-SF mediated joint destruction [206]. Likewise, it has been described that VIP counteracts the stimulatory effect of pro-inflammatory mediators, including TLR3 and TLR4 ligands, TNF-α, and IL-17, on the expression of IL-17 receptors and the IL-12 family of cytokines in RA-SF, which, in turn, mediates their cross-talk with Th1/Th17 cells [207]. Along with the anti-inflammatory effects of VIP in RA-SF through its action on TLR, this neuropeptide has been described to decrease the pro-inflammatory peptides corticotropin releasing factor (CRF) and urocortin (UCN)-1, while increasing the expression of the potential anti-inflammatory agents UCN-3 and CRF receptor 2 (CRFR2). Moreover, VIP is able to inhibit CREB activation, cyclooxygenase 2 expression, and prostaglandin 2 (PGE2) secretion in RA-SF [208]. 

In line with these findings, the potent anti-inflammatory role for VIP on cellular components of the immune system in the context of RA has also been validated by in vitro studies [158]. Upregulated levels of pro-inflammatory mediators, including TNF-α, IL-6, and CXCL8 and CCL2 chemokines, in polyclonally stimulated peripheral blood lymphocytes from RA patients were reduced after treatment with VIP [167]. Furthermore, regarding its effects on macrophages, VIP was able to impair the acquisition of the pro-inflammatory polarization profile described for macrophages in RA synovium, favoring instead an anti-inflammatory phenotype [32]. Additionally, the involvement of VIP in the modulation of Th subsets has been extensively studied, as previously detailed. 

Apart from the effects of VIP treatment in animal models and in cultured cells from RA patients, recent studies have focused on evaluating the potential value of endogenous VIP as a biomarker in RA, as we discuss later. 

#### 5.1.2. VIP in Osteoarthritis

OA is a chronic rheumatic disease and is considered the most prevalent in developed countries and the main cause of incapacity in the elderly population. It is a complex multifactorial disease and is the clinical endpoint of heterogeneous disorders with common clinical, pathological, and radiological characteristics, resulting in the alteration of one or more joints [209,210,211,212,213]. Although it is usually an age-related disease, OA is also associated with other multiple risk factors that culminate in joint dysfunction, including genetic predisposition, epigenetic factors, gender, obesity, exercise, work-related injury, and trauma [209,214,215,216].

OA is characterized by cell stress and extracellular-matrix (ECM) degradation, resulting in an imbalance in joint-tissue metabolism, which culminates in a progressive loss of synovial joint function, with pain and disability. While cartilage degradation is the main event, the view of OA as solely a pathology of cartilage has changed in recent years. This pathology affects the whole joint, resulting in the remodeling of adjacent subchondral bone, osteophyte formation, and synovial inflammation [213,217,218,219,220,221,222,223,224,225,226]. Although OA has an important mechanical component, it is currently also considered as a low-grade inflammatory disease. The biological imbalance and the mechanical stress lead to a pathological situation, with altered chondrocyte behavior, which results in the release of inflammatory mediators and ECM-degrading enzymes [19,20,21,22]. All of these factors, along with the inhibition of cartilage biosynthesis, increase the fragility and loss of cartilage integrity [23]. Although synovitis is usually localized and may be asymptomatic in OA [24], synovial activation causes the release of inflammatory mediators and proteases that accelerate the progression of the disease [2,25,26]. Moreover, the subchondral bone is also affected and is involved in the progression of OA through the release of catabolic mediators that promote an altered metabolism in chondrocytes [14,27].

The majority of available therapies for OA focus on relieving symptoms rather than slowing the progression of the disease. Therefore, it is important to find new therapeutic targets for the development of new drugs to treat the disease [227,228]. 

While the association of VIP with RA has been widely studied, its role in OA is not well established, although it is the second rheumatic pathology in which more advances have been obtained in the study of the VIP function [229] (Figure 3). Less is known about the role of VIP in other disorders such as systemic lupus erythematosus or spondyloarthritis (SpA). The effects of VIP reported in rheumatic diseases could be mediated in part by its action on the SF, as has been described in several in vitro studies [82,98,142,207,230]. OA-SF expresses and releases VIP, with a greater expression than RA-SF [38]. However, its expression is decreased in the synovial fluid and cartilage of OA patients compared to healthy controls, which could contribute to the pathology [231,232]. Regarding VIP receptors, both VPACs are detected in OA-SF with a greater expression of VPAC1. Pro-inflammatory mediators released to the joint microenvironment during the disease, such as TNF-α, decrease the expression of VIP, and modulate the VPAC1/VPAC2 ratio, therefore approaching its profile to that of RA-SF [38].

VIP is able to counteract the action of pro-inflammatory mediators, alleviating the inflammation and the pain in OA. VIP reduces the serum levels of TNF-α and IL-2 and increases serum IL-4 in a rat model of knee OA. In this model, VIP also inhibits proliferation of OA-SF and decreases the production of TNF-α, IL-2, MMP-13, and ADAMTS-5 (a disintegrin and metalloproteinase with thrombospondin motifs-5), at the same time that it induces the expression of type II collagen and osteoprotegerin, by inhibition of NFκB signaling [232,233]. In addition, VIP modulates the corticotropin-releasing factor family of neuropeptides, also increasing the expression of the potential anti-inflammatory mediators UCN-2 and -3, as well as CRFR2 in OA-SF. Moreover, VIP increases cAMP and induces CREB activation in OA-SF [208], which would support its anti-inflammatory role through the inhibition of other signaling pathways, involving JNK-MAPK, NFκB, or c-Jun, inhibiting the production of pro-inflammatory mediators and promoting the expression of anti-inflammatory cytokines [85,234,235,236]. 

On the other hand, some studies reported that the accumulation of VIP in joints can also contribute to the pathogenesis of OA. Thus, VIP treatment in rat OA knees promotes synovial hyperemia, as well as sensitization of joint afferent fibers via AC/cAMP/PKA, also increasing firing rate and decreasing mechanical threshold during movement. Therefore, VIP might promote mechanosensitivity and pain in rat OA models [37,225,232,237,238]. Moreover, Rahman et al. reported that VIP stimulates PGE2 production in human articular chondrocytes, human osteoblast-like cells, and human SF, as well as cAMP production in human osteoblast-like cells, suggesting a pro-inflammatory role for this peptide [239]. Another study also related increased VIP levels in the synovial fluid to the presence of synovitis in OA patients [240], suggesting that both downregulation and upregulation of VIP could contribute to the OA pathology [232].

In addition to the inflammatory process, ECM degradation and cartilage loss is a key factor in the OA pathology. In this regard, VIP might prevent cartilage damage, since VIP modulates the profile of ECM-degrading enzymes released to the joint microenvironment by SF from OA patients. Thus, VIP decreases the expression and activity of the proteinase urokinase-type plasminogen activator (uPA), as well as the production of its receptor (uPAR), after stimulation with the pro-inflammatory cytokine IL-1β or the degradative mediators 45kDa fibronectin-fragments (Fn-fs). On the other hand, VIP induces the production of the plasminogen activator inhibitor-1 (PAI-1) under basal conditions in these cells. Furthermore, VIP reduces the production of MMP-9 in IL-1β- or Fn-fs-stimulated OA-SF, as well as MMP-13, the main proteinase involved in the degradation of type II collagen, after stimulation with Fn-fs [220]. In addition, VIP decreases the production of ADAMTS in OA-SF, including the aggrecanases ADAMTS-4 and -5, key proteinases in the degradation of aggrecan from the cartilage ECM, after IL-1β or Fn-fs stimulation, as well as the cartilage oligomeric matrix protein (COMP)-degrading ADAMTS, ADAMTS-7 after both stimuli, and -12 after Fn-fs treatment. In this sense, VIP also reduces COMP degradation from cartilage explants cultured with IL-1β- or Fn-fs-stimulated OA-SF, as well as the aggrecanase activity and glycosaminoglycans (GAGs) release only after Fn-fs stimulation. Moreover, VIP inhibits the activation of Runx2 transcription factor and Wnt/β-catenin signaling involved in ECM remodeling and proteinase expression, after the stimulation of these cells with both stimuli [241].

Few studies have focused on the presence of VIP and its receptors in chondrocytes [37]. As previously described in SF, articular cartilage from OA patients also has lower VIP levels compared to controls. Moreover, VIP expression in synovial fluid is positively correlated to its optical density in articular cartilage [231]. Juhász et al. described the expression of VPAC1, VPAC2, and PAC1 in chicken chondrogenic cells [242,243]. 

Concerning subchondral bone, VIP receptors have been described on osteoclasts and osteoblasts of several species, including human, mouse, and rat [37,244]. VIP inhibits osteoclast-mediated bone resorption and induces the production of IL-6 from osteoblasts, regulates the expression of osteoclastogenic factors like RANKL and OPG in osteoblasts, and seems to be involved in osteoblastogenesis [37,242,245,246]. In addition, VIP promotes osteoblast activity and proliferation and stimulates bone remodeling [244,247]. Furthermore, Xiao et al. showed higher VIP levels in the femoral bone from OA postmenopausal women compared to those with osteoporosis, where VIP was also positively associated with pain [248].

Recent studies also associate VIP levels to the progression of rheumatic diseases. In this regard, VIP levels in synovial fluid and cartilage of OA patients are negatively associated with progressive joint damage, being a potential indicator of disease severity [231]. In addition, VIP could be postulated as a potential therapeutic target in OA, since it is involved in the activation of several anabolic signaling pathways in the synovial joint [245].

### 5.2. Inflammatory Bowel Disease

Inflammatory bowel disease is the prevailing gut autoimmune disorder and comprises Crohn’s disease and ulcerative colitis (UC). As with other autoimmune diseases, the origin is multifactorial and comprises genetic, environmental, and host-related factors that affect the development of bowel inflammation [249].

UC lesions are located within the colon, while CD is a relapsing remitting granulomatous disease, which can affect any segment of the digestive tract, producing transmural lesions. Although IBD pathogenesis is unclear, an atypical immune response to intestinal microbial products and/or food allergens represents an important causal factor. Moreover, interactions between the enteric nervous system (ENS) and the immune system play an important role in its pathophysiology [250,251]. These communications include the secretion of neuropeptides, which conduct signals bidirectionally between enteric neurons and immune effectors [252]. VIP and its receptors are expressed in the gastrointestinal tract to perform its anti-inflammatory/immunomodulatory action. The source of endogenous VIP in the gut could be of nervous origin, or from lymphoid cells. Concerning receptors, as we previously described, VPAC receptors are expressed in monocytes, macrophages, and T and B cells, as well as in myeloid cells, such as mast and polymorphonuclear cells. 

To date, different models of chemically induced IBD have been characterized, showing several clinical, histological, and immune-response characteristic of UC and CD: the Dextran Sodium Sulfate (DSS), the oxazolone-induced colitis, TNBS, and Dinitrobenzene sulfonic acid (DNBS). The murine model has benefits, as well as limitations, in some characteristics of their clinical, immunological, and histopathological relevance to IBD. Administration of 3–10% DSS in the drinking water is one of the most common chemical methods used to induce colitis in rodents [253]. The oxazolone-induced colitis represents a model of Th2-driven inflammation. In this model, colitis is induced by intracolonic instillation of the haptenating agent oxazolone dissolved in ethanol after a skin pre-sensitization step [254]. However, there is limited information about the time course and cytokine profile of the immune response involved in this model of colitis.

Other models of colitis include the hapten-induced DNBS or TNBS that are administered by rectal instillation diluted in ethanol. The haptenization of host proteins induced infiltration of neutrophils, macrophages, and Th1 lymphocytes in the injured mucosa. In comparison to DNBS, TNBS is considered to be a hazardous chemical due to its highly oxidative properties. Since it was developed more recently, research using the DNBS-induced model is less common [255].

Gut inflammation and a differential expression profile of cytokines are key properties of their immune response. The TNBS colitis model develops with elevated Th1–Th17 response (increased IL-12 and IL-17), while DSS colitis switches from a Th1–Th17-mediated acute inflammation (increased TNF-α, IL6, and IL-17) to a central Th2-mediated inflammatory response (increment in IL-4 and IL-10 and associated reduction in TNF-α, IL6, and IL-17) [256]. This dissimilar cytokine profile has been used to establish an equivalence with human IBD. Thus, TNBS colitis mimics CD, while chronic DSS-colitis mimics UC [257]. 

The first report on the role of VIP as a therapeutic agent in IBD was published in 2003, in the TNBS colitis model [7]. VIP treatment reduced the clinical and histopathologic severity of TNBS-induced colitis, abolishing body-weight loss, diarrhea, and macroscopic and microscopic gut inflammation. The VIP effect is mediated by both innate and acquired immune responses. Regarding innate immunity, administration of VIP in the TNBS model decreased myeloperoxidase activity in colon extracts, a specific marker of neutrophils, and reduced the expression of receptors involved in neutrophil recruitment, such as CXCR1 and CXCR2 [7,156,258]. CD4^+^ T helper cells are major initiators of IBD. CD4^+^ T cells are enriched in the gut of patients with CD and UC and blockade or reduction of CD4^+^ T is effective in treating patients with IBD [259].

Th1 and Th17 subsets are important players in the development of CD [260]. In the TNBS model, we reported that VIP reduced IFN-γ and TNF-α enhancing IL-4 and IL-10 levels in colon and cell cultures from splenocytes and lamina propria immune cells, thus promoting Th2 vs Th1 responses. VIP diminished IL-17, IL-21, and IL-17R mRNA expression in the colon, supporting an inhibitory action over the Th17 cell subpopulation. Interestingly, VIP increased the Foxp3 and transforming growth factor (TGF)-β mRNA expression in CD4^+^ cells from mesenteric lymph nodes, as well as the IL-10 expression in the colon upregulating Treg responses [7,156,258]. Finally, VIP also reduced the TNBS-induced numbers of TCD4 lymphocytes, whereas it induced an increase in the number of B-lymphocytes (CD19^+^) in mesenteric lymph nodes.

To date, different mechanisms involved in the therapeutic effect of VIP have been described. One of the first actions described was the VIP modulation of TLR. Among the environmental factors, the modification of gut microbiota or dysbiosis has been reported as a key element in the development of IBD [261].

Additionally, the receptors of the innate immune system, TLRs, affect many aspects of IBD etiology, including immune responses and microbiota. Differential expression of TLRs in IBD patients in comparison with healthy donors has been characterized. Modification of TLR expression or signaling has been reported, not only in experimental models of IBD in mice, but also in human IBD. Most TLR signaling pathways participate in the development of IBD and are sometimes beneficial and other times harmful [262]. Nevertheless, much of the evidence has indicated that the TLR2 and TLR4 signaling pathway has a negative role in IBD. It was reported that the inhibition of TLR2–TLR6/1 activity ameliorated DSS-induced colitis. In healthy patients, TLR4 is expressed at a low level in intestinal epithelial cells; however, its expression was upregulated in the intestinal epithelia of patients with active UC, suggesting that TLR4 could be involved in UC disease development [262].

In the TNBS-induced colitis model, VIP treatment exerts a time-course inhibition of TLR2 and TLR4 expression in colon epithelial and mononuclear cells. Moreover, VIP acts at a systemic level in lymph nodes. Mesenteric lymph nodes are the draining nodes of the intestinal tract that regulate the traffic of lymphoid cells. VIP inhibits the TNBS-induced TLR2 and TLR4 overexpression in macrophages, dendritic cells and the lymphocyte subpopulations, T CD4^+^, T CD8^+^, and B CD19^+^ [134,135]. The peptide also enhances the expression of Foxp3 and TGF-β, which are both involved in regulatory T-cell function. Overall, we reported that, after specific stimulation of TLR2 and TLR4, VIP exerts homeostatic function, balancing innate and adaptive immune responses in the murine model of CD, both locally in the colon and at the periphery in lymphoid nodes [131,156,263,264].

Another study using the TNBS-induced colitis model reported that treatment with VIP did not modify the clinical and histological parameters [265]. However, another recent study confirmed our results. Because the nanocarrier sterically stabilized micelles (SSM) protect peptides from enzymatic degradation, ameliorating their bioavailability and half-life, Jayawardena et al. developed sterically stabilized micelles of VIP (VIP-SSM). They characterized the healthy role of VIP and VIP-SSM in the DSS-induced colitis model. At clinical and histological levels, VIP and nanoparticles of VIP treatment decreased the pro-inflammatory cytokine profile in the colon, reducing tight junction and ion-transporter protein expression associated with severe DSS colitis [266]. 

It is also important to note that VIP has shown beneficial effects in other models of colitis, such as colitis induced by *Citrobacter rodentium* [267] and the oxazolone-induced colitis [268].

Results are variable in knockout mice of the VIP/VPAC receptor axis, [269]. The VPAC2 receptor KO mice showed worse progression of DSS-induced colitis, whereas VPAC1 knockout DSS-induced colitis in VPAC1-KO mice was resistant to colitis [270]. Concerning VIP knockout mice, the results using the chemical-induced colitis models are contradictory. Thus, DNBS and DSS-induced colitis were more severe in VIP-KO than wild-type mice. VIP treatment recovered the phenotype, protecting VIP-KO mice against DSS colitis. Moreover, VIP is beneficial for the development and maintenance of a colonic epithelial barrier structure under physiological conditions and promoting epithelial repair and homeostasis during colitis [271]. Abad et al. reported in the TNBS model that mice lacking VIP developed reduced colitis [272]. These discordant results using the KO model of the VIP axis could be explained by the presence of differential microbiota, by alterations in the development of the chemical-induced model of colitis or by the existence of compensatory mechanisms in VIP-KO, by PACAP, a related peptide, or by another mediator.

Despite the scarce dissimilar results about the effect of VIP in the TNBS model of CD and those obtained with the KO mice, the conclusive results are robust and are summarized in Figure 4.

Data on the role of VIP in humans are scarce and are relative to the presence of the peptide in health conditions and in IBD patients. Contradictory results about alterations in gut VIP innervation in IBD patients have been reported. Several studies described enhanced VIP expression in the intestine in IBD patients [273]. Moreover, an increase in VIP immunoreactivity both in nerve fibers and neurons were characterized in CD patients [264]. Conversely, other studies have characterized a reduction in the abundance of VIP-immunoreactive nerve fibers in the lamina propria and submucosa in both CD and UC patients. Remarkably, the variation in the decrease was significantly related to the severity of the disease [274]. In general, it is well recognized that a broad loss of mucosal neuropeptidic innervation may be related to areas of high inflammation; thus, the contradictory results could be explained by the experimental conditions.

In a recent elegant study, Sun et al. described the beneficial role of VIP in UC patients. In agreement with data reported in several rheumatic diseases [245,275,276], they found that serum VIP levels are lower in UC patients than in healthy controls. The study provided evidence showing that VIP serum levels are lower in IgE^+^ UC patients than that in IgE¯ UC patients [268].

The same authors found that in the regulatory B cells (Bregs) from peripheral blood of UC patients, immune suppressive function is impaired, probably due to lower serum VIP levels and lower IL-10 expression in Bregs. This expression increases with the presence of VIP, which stabilizes IL-10 expression in Bregs. In brief, they demonstrated that VIP administration restored Breg function, inhibited pro-inflammatory cytokine production, prevented the allergen-specific T-cell response, and reestablished colon tissue structure in experimental colitis. All of these results suggest that VIP is a potential therapeutic agent for UC patients with atypical immune responses to food allergens [268].

To date, no therapy is yet available for the treatment of IBD and combined therapy seems to be the best approach [261]. Despite the contradictory data from KO models, the VIP axis represents a promising candidate for use in combined therapy due to its multistep action on the immune response.

### 5.3. Central Nervous System Diseases

The involvement of inflammatory processes and the adaptive immune system in the pathophysiology of neurodegenerative diseases is supported by evidence from a variety of studies [277,278,279,280]. Thus, inflammatory responses may be involved in both regenerative and degenerative processes, e.g., in multiple sclerosis and Parkinson’s disease (PD). During neuroinflammation, the activation of the glial cells of the brain, mainly microglia and astrocytes, release several factors, many of which are pro-inflammatory, neurotoxic, and damaging to nerve cells.

In vitro and in vivo investigations have described potent neuroprotective features for VIP promoting neural cell proliferation, survival, axon regeneration, and production of neurotrophic factors, as well as inhibition of inflammation [108,281,282]. All of this indicates that the VIP/receptors axis could be a novel therapeutic target in multiple sclerosis and Parkinson’s disease.

Multiple sclerosis is a chronic inflammatory autoimmune and neurodegenerative pathology of the CNS that leads to demyelination. Experimental autoimmune encephalomyelitis is the most common animal model for MS sharing many clinical and pathophysiological features resulting in the generation of autoreactive T-cells, eventually culminating in myelin destruction. Like that observed in other inflammatory diseases, treatment with VIP reduced the clinical and pathological scores in EAE with a blockade of symptoms lasting 60 days. These effects were associated with decreased spinal cord levels of pro-inflammatory cytokines (TNF-α, IL-6, IL-1β, IL-18, and IL-12), iNOS, chemokines (CCL5, CCL3, CXCL1, CCL2, and CXCL10), and CC chemokine receptors (CCR-1, CCR-2, and CCR-5) and increased levels of the anti-inflammatory cytokines IL-10, IL-1Ra, and TGF-β [180].

These investigations revealed that VIP treatment decreases the presence of encephalitogenic Th1 cells in the periphery and the CNS. As a consequence, VIP reduces the appearance of inflammatory infiltrates in the CNS, the loss of oligodendrocytes, and the subsequent demyelination and axonal damage typical of EAE [283].

Several studies point to Tregs as key agents in MS and EAE controlling self-reactive cells and inducing the decrease in inflammation [284,285,286,287]. Administration of VIP to EAE mice induces the expansion of Treg cells that express CD4^+^ CD25^+^ Foxp3 and produces IL-10/TGF-β in the periphery and the CNS [288].

In accordance with these results, it has been described that mice with a genetic deletion of the VPAC2 gene exhibit an exacerbation of EAE induced by MOG35-55 compared to wild-type mice, presenting an increased pro-inflammatory cytokine profile (TNF-α, IL-6, IFN-γ, and IL-17) and reduced production of anti-inflammatory cytokines (IL-10, TGF-β, and IL-4) in the CNS and lymph nodes. In addition, the proliferative index and the in vivo suppressor activity of CD4^+^ CD25^+^ FoxP3^+^ Tregs are markedly reduced in KO VPAC2 mice with EAE [289]. These results point toward an important protective role for the VPAC2 receptor against autoimmunity and as an anti-inflammatory mediator [289].

Unexpectedly, and in contrast, Abad et al. found that VIP-KO mice are highly resistant to EAE. This finding was confirmed by histopathology and clinical evaluation. Supporting this phenotype, the levels of multiple pro-inflammatory cytokines in the spinal cord were strikingly reduced in the KO. The authors found that immune cells were trapped in the meninges of the brain and spinal cords and failed to invade the CNS parenchyma, suggesting a defect in immune-cell migration [290]. Clinical disease in these mice was blocked at a step downstream from immunization. Similarly, EAE clinical symptoms are significantly ameliorated in VPAC1-KO mice. The results demonstrate stronger Th1 and Th17 responses, which are known to induce the pathogenesis of EAE, but reduced Th2 responses in these mice. As the phenotype of VPAC1 is opposite that of VPAC2-KO mice, it has been suggested that, in addition to Th polarization, other events are differentially mediated by VPAC1 and VPAC2, and this may depend on different factors, such as their level of expression, the state of cellular activation, the interaction with other mediators present in the microenvironment, such as inflammatory factors, and finally, the disease phase [291].

Studies in patients with MS have reported alterations in components of the VIP/receptors signaling system. Andersen et al. found a reduced VIP immunoreactivity in the cerebrospinal fluid of patients diagnosed with MS [292]. Similarly, Baranowska-Bik et al. observed a tendency toward reduced levels of VIP in the cerebrospinal fluid of multiple sclerosis patients, although this difference was not statistically significant [293]. Interestingly, CD4^+^ cells derived from the peripheral blood of patients with MS show a differential expression of the VPAC1 and VPAC2 receptors, compared to those from healthy controls, depending on the activation status of the cells. Without stimulation, similar patterns of VIP receptor expression are detected in CD4^+^ cells of subjects with MS and healthy controls with a visible expression of VPAC1 and minimum levels of VPAC2. However, the markedly decreased expression of VPAC1 observed after stimulation and activation of CD4^+^ T cells is compensated for by a higher expression of VPAC2 in healthy individuals. Nevertheless, activated CD4^+^ T cells from SM patients exhibit an altered expression of VPAC2 as a result of altered gene regulation in the promoter region of the VIP receptor. As a consequence, CD4^+^ T cells were less sensitive to VIP and biased the system predominantly in a Th1 direction [294].

Finally, the role of the VIP/VPAC axis has also been investigated in Parkinson’s disease. In this progressive degenerative movement disorder, the key roles played by the subsets of CD4^+^ cells, especially the Treg whose number or activity is reduced in this pathology, have recently been highlighted. This, together with microglial activation, leads to changes in the microenvironment of the affected brain with oxidative stress, inflammation, and defective protein folding [295,296,297].

Using murine models of Parkinson’s disease, administration of VPAC2 agonists has been shown to increase Treg activity without altering cell numbers, reduce microglial inflammatory responses, increase survival of dopaminergic neurons, and improve striatal densities [63,296]. 

### 5.4. Other Autoimmune Disorders

Type 1 diabetes and Sjögren’s syndrome (SS) represent other autoimmune diseases in which the beneficial effects of VIP have been shown. Type I diabetes is an autoimmune disease mediated by T cells associated with the overexpression of inflammatory mediators and the alteration of different subsets of T cells that attack the insulin-producing cells in the pancreas. 

Several animal models that develop spontaneous type 1 diabetes have been described, such as NOD mice that exhibit T-cell-mediated insulitis linked to the genes of the major histocompatibility complex. In the NOD mouse model, VIP prevents the increase in the proportion of Th1 to Th17 cells, changes the Tregs/Th17 ratio that leads to tolerance, and reverses the proportion of subsets of Th1/Th2 cells associated with autoimmune pathology. These effects add to the decrease in pro-inflammatory mediators, resulting in a reduction in the destruction of β cells in the pancreas [10].

Studies in KO mice have confirmed the importance of the VIP/VPAC axis in the functionality of the endocrine pancreas. Thus, Martin et al. found that VIP-KO mice exhibit elevated plasma glucose, insulin, and leptin levels [298]. In VPAC2-KO mice, glucose-induced insulin secretion is decreased, with no change in glucose tolerance and mice deficient in VPAC1 show small dysmorphic islets of Langerhans and exhibit impaired neonatal growth that leads to intestinal obstruction and hypoglycemia [299]. Moreover, selective overexpression of the human VIP gene increases glucose-induced insulin secretion in pancreatic β-cells and ameliorates glucose intolerance of 70% depancreatized mice [300].

SS is an autoimmune disease characterized by the infiltration of T lymphocytes at the level of the salivary and lacrimal glands that causes their destruction and the appearance of symptoms related to dry mucous membranes. Recently, the effects of VIP on the immune response and secretory function of the submandibular glands have been investigated by using the NOD model of SS, which develops secretory dysfunction and early loss of glandular homeostatic mechanisms, with mild infiltration in the glands.

Li et al. showed that VIP treatment was able to reduce immune lesions in the exocrine glands and improve the secretory function of these glands by negatively regulating the expression of IL-17A in the exocrine glands. It also improved the secretory function of the exocrine glands by increasing the expression of AQP5, a protein that participates in the transport of water through the glandular epithelium [301].

In the course of salivary function impairment in the NOD mouse model, a progressive decrease in VIP expression in the submandibular glands is observed compared to normal mice. The loss of endogenous VIP is associated with a loss of acinar cells through apoptotic mechanisms that could be further induced by TNF-α and reversed by VIP through a PKA-mediated pathway. The clearance of apoptotic acinar cells by macrophages is impaired by NOD macrophages contributing to the loss of gland homeostasis [302].

Lodde et al. constructed the vector recombinant serotype 2 adeno-associated virus, encoding the human VIP transgene (rAAV2hVIP), to explore its usefulness in SS management. Instillation of rAAV2hVIP in the submandibular glands of NOD mice leads to higher salivary flow rates and increased expression of VIP in the glands and serum, as well as to a reduction of cytokines IL-2, IL-10, and IL-12 (p70) and TNF-α in SMG extracts, and of serum CCL5, compared to the control vector. This work indicates that VIP may be a promising agent for the treatment of the salivary component of SS [9].

Finally, data from humans reveal that monocytes from SS patients show increased expression of VPAC2, which is absent in the monocytes of normal subjects without changes in the expression of VPAC1. This altered expression correlates with an impaired phagocytosis of apoptotic epithelial cells, with reduced engulfment capacity and failure to express an immunosuppressant cytokine profile that is not restored by VIP. This differential expression of VPAC2 associated with phagocytic dysfunction suggests its potential as a functional biomarker in SS [303].

## 6. VIP as a Therapeutic Agent: Limitations and Perspectives

The treatment of inflammatory and autoimmune diseases is a challenge. Advances in knowledge of the underlying pathophysiological mechanisms, as well as the discovery of biological therapies against potent mediators of inflammation, have revolutionized the way these diseases are treated. Newly developed molecules aim to diminish the impact of these diseases on the quality of life of patients, although, to date, there is no curative treatment.

Since the discovery of VIP half a century ago, more knowledge about its biology, its signaling mechanisms, and its powerful anti-inflammatory effects, as well as its immunoregulatory capacity, have made it a potential therapeutic agent for diverse diseases. Among these disease are asthma [304], pulmonary hypertension [305], sarcoidosis [306], neurological diseases such as Alzheimer’s and Parkinson’s [307,308], inflammatory bowel diseases such as Crohn’s [7,264], autoimmune diabetes [10], and cancer [309].

Marketed under the name Aviptadil, VIP has been used in the clinic successfully in the treatment of pulmonary hypertension and sarcoidosis. However, the potential of this peptide at the therapeutic level in clinical practice is still far from its theoretical potential. This is due to its high sensitivity to degradation by proteases, spontaneous hydrolysis, and catalytic antibodies [310,311]. A second limitation for the use of VIP in humans is due to cross-interactions, given their ability to bind to different GPCRs, their functional pleiotropism, and their ubiquity. In addition, systemic administration of VIP with binding to multiple cell targets with high affinity could cause unwanted adverse effects [312,313].

Therefore, strategies have been developed to overcome these difficulties. Thus, distribution systems directed against specific targets that also protect the peptide against its degradation are desirable options. Recent advances in this field include the following: the use of metal nanoparticles, which seem to increase the therapeutic potential of VIP both in terms of target and distribution [314]; the use of modified liposomes with lipopeptides conjugated with VIP, which have demonstrated a selective recognition of VPACs and a more effective antitumor activity in a recent study with human osteosarcoma lines [315], as well as nanomicelles, tested in breast cancer [316].

Interestingly, a single subcutaneous injection of a low dose of camptothecin sterically stabilized micelles conjugated with VIP (CPT-SSM-VIP) administered to mice with collagen-induced arthritis was able to abrogate joint inflammation, with no apparent systemic toxicity and with similar efficacy and safety compared to methotrexate, used clinically for RA treatment [317]. The efficacy of this therapeutic approach has been confirmed in the murine model of colitis induced by DSS. Similarly, the anti-inflammatory and antidiarrheal effects of VIP can be achieved effectively when administered as a nanomedicine.

In relation to the problem posed by VIP’s binding to different receptors, stable analogues of the VPAC1 and VPAC2 receptors have been developed recently, based on the technology of the peptidases-resistant foldamers (Longevity Biotech), LBT3627 and LBT3393. LBT-3627 is a VPAC2 selective neuroprotective agent that has been successfully investigated in the preclinical phase in a Parkinson’s model (Olson et al. 2015 [63]. The results obtained highlight the therapeutic immunomodulatory potential of this agonist to restore Treg activity, attenuate neuroinflammation, and intercept dopaminergic neurodegeneration in PD, as we mentioned above [296].

Finally, gene therapy with VIP, using lentiviral vectors, has yielded good results in the CIA model [318], and VIP adenoviral vectors have also been developed [9]. However, these approaches continue to lack cellular and tissue specificity. Thus, another possibility under study is cell therapy with dendritic cells transduced with a VIP lentiviral vector (LentiVIP-CDs), whose therapeutic effects in sepsis and EAE models have been very positive with a single local administration [179].

Looking toward the future, despite advances in therapeutic options, there is still a need to continue researching the design and transfer to the clinic of stable VIP analogues and specific VPAC1- and VPAC2-receptor drugs, directed against specific objectives, as well as biomarker field approaches to intervene earlier in the course of the disease.

## 7. Potential of VIP Axis as a Biomarker for Personalized Treatment in Rheumatic Diseases

In addition to its potential use as a therapeutic agent, the VIP axis could be used in a second potential translational strategy as a prognostic biomarker. Different studies have described an altered expression in the VIP/VPAC axis in autoimmune diseases and in the modulation of the inflammatory immune response in rheumatic diseases [82,98,153,167]. In juvenile idiopathic arthritis, serum VIP levels are decreased in patients who manifest more disease activity characterized by cardiac autonomic neuropathy associated with parasympathetic dysfunction [319]. In RA, the expression of VPAC1 is decreased in peripheral blood mononuclear cells (PBMCs) [318], and a lower expression of VPAC1 mRNA is also observed in patients with early arthritis (EA) [320]. However, the expression of VPAC2 is increased in SF and PBMCs show an increased expression of VPAC2 mRNA [38,320]. A deregulated expression of VPAC2 has also been described in monocytes isolated from patients with SS and in activated CD4^+^ T cells from patients with multiple sclerosis [294].

The abnormal expression of the VIP axis in autoimmune diseases directed the investigations toward the study of its association with the clinical course of some of these diseases and their possible prognostic value. In RA, early diagnosis and the establishment of immediate and effective therapy are essential to prevent greater disease severity [321,322]. Although different parameters have been proposed as prognostic markers for RA (such as rheumatoid factor ACPAs, erythrocyte sedimentation rate, and C-reactive protein), they are only capable of classifying 65% of patients [323,324,325,326]. In this sense, patients with RA with high or moderate activity after two years of follow-up had lower levels of VIP at baseline [275]. In multivariate analyses, it was observed that ACPA-negative patients had an odds ratio (a statistic that quantifies the strength of the association between two events) of 6.1, having high activity at two years of follow-up if their initial serum VIP levels were low. This allows the classification of a group of patients with a greater need for treatment within the ACPA seronegative RA patients. Another factor that makes VIP a potential prognostic marker worthy of further study is the fact that several single nucleotide polymorphisms (SNPs) in the VIP gene are associated with differences in serum VIP levels in patients with EA. The combination of three SNPs (minor alleles in rs688136 and absence of minor alleles in rs35643203 and rs12201140) in the VIP gene allows the identification of patients with less-severe disease, and thus possibly good candidates for less-intensive therapy [327].

Regarding the VIP receptors, it has been observed that the expression of VPAC1 and VPAC2 could reflect the clinical status in patients with EA with a significantly lower expression of VPAC1 when patients have systemic inflammatory activation characterized by high serum levels of IL6 and higher levels of Disease Activity Score 28 (DAS28). DAS28 is an index of the disease activity developed and validated by the EULAR (European League Against Rheumatism) [320]. In addition, the VPAC2 expression prevailed over VPAC1 in cells polarized toward Th17 of EA patients [170]. VPAC2 can also mediate anti-inflammatory effects when the expression of VPAC1 is low [38].

Serum VIP levels also showed a prognostic value in spondyloarthritis, a family of rheumatic diseases that share clinical and radiological manifestations where the most prevalent group is ankylosing spondylitis. These patients are HLA-B27 positive, and their inflammation usually occurs with enthesitis and bone formation that can lead to ankylosis. Patients with SpA presented a wide heterogeneity in terms of clinical manifestations, and there are no good biomarkers that predict progression. In early SpA, patients with lower VIP levels showed more disability and factors related to increased inflammation (bone edema on MRI scan, anemia, enthesitis, and cutaneous psoriasis) [276]. Finally, it has been described that VIP levels may have a protective role in the progression of OA [231].

In summary, VIP is an excellent aspirant to be used in clinical practice as a prognostic biomarker that would complement existing markers, such as ACPAs. Concerning receptors, they emerge as good candidates for activity biomarkers, and current studies would expand their potential as a severity biomarker.

Figure 5 summarizes the current advances in the role of the VIP/receptors axis as biomarkers in rheumatic diseases.

## 8. Conclusions

On the 50th anniversary of VIP’s discovery, this review updates our knowledge about the regulatory functions of the VIP/receptors axis in the immune system and presents a spectrum of potential clinical benefits applied to inflammatory and autoimmune diseases. This article gathers the findings and advances achieved in this field, thanks to the work of numerous researchers, from both basic and translational research areas.

Recent progress in improving the stability, selectivity, and effectiveness of VIP/receptors analogues and new routes of administration are highlighted, as well as important advances in their use as biomarkers, contributing to their potential application in precision medicine.

Despite the achievements, it is necessary to continue researching the design of analogue drugs that are stable, safe, and directed against specific objectives and in the validation of the VIP/receptors axis as biomarkers such that their application in clinical practice becomes a reality for our Very Important Patients.

## Figures and Tables

**Figure 1 ijms-21-00065-f001:**
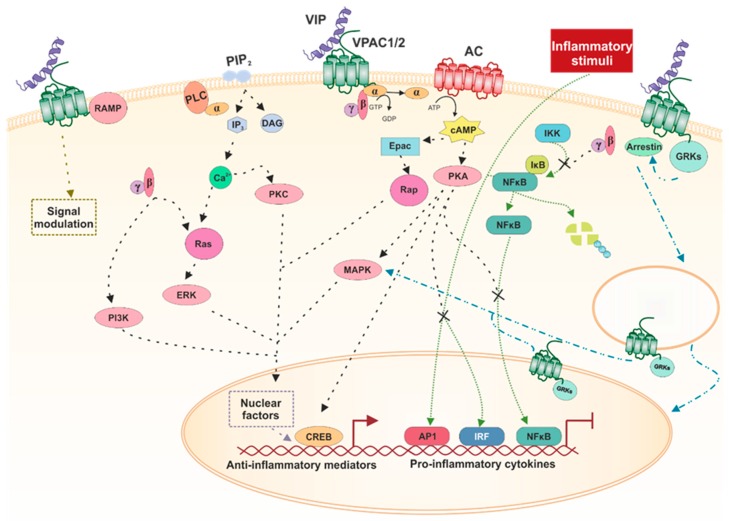
VIP/VPAC receptors’ axis signaling pathways. VIP/VPAC receptors’ binding activates a cAMP-dependent signaling pathway mediated by the induction of AC (black arrows). Then, cAMP activates PKA, which in turn induces nuclear translocation of CREB (black arrows). Besides, PKA inhibits the activation of pro-inflammatory transcription factors such as AP-1, IRF, or NFκB (black cross). Additionally, cAMP in a PKA-independent way simulates EPAC (black arrows). This second messenger induces anti-inflammatory transcription factors. VPAC receptors can also activate PLC and PI3K (black arrows). Both signaling pathways produce the nuclear translocation of anti-inflammatory transcription factors. These receptors can also interact with accessory RAMPs, modulating the canonical signaling pathways (dark green arrow). Furthermore, VPAC1 is able to translocate to the nucleus by the interaction with GRKs (blue arrows). Inflammatory stimuli activate signaling pathways (light green arrows).

**Figure 2 ijms-21-00065-f002:**
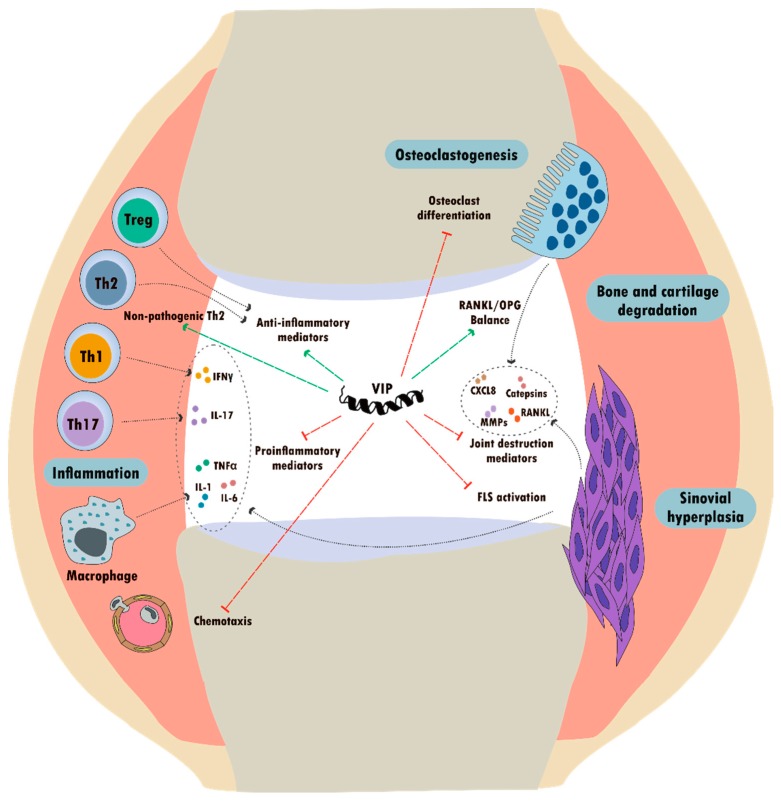
Biological effects of VIP in rheumatoid arthritis. Schematic representation of an RA joint. Green arrows indicate “induction”, whereas red arrows indicate “inhibition”.

**Figure 3 ijms-21-00065-f003:**
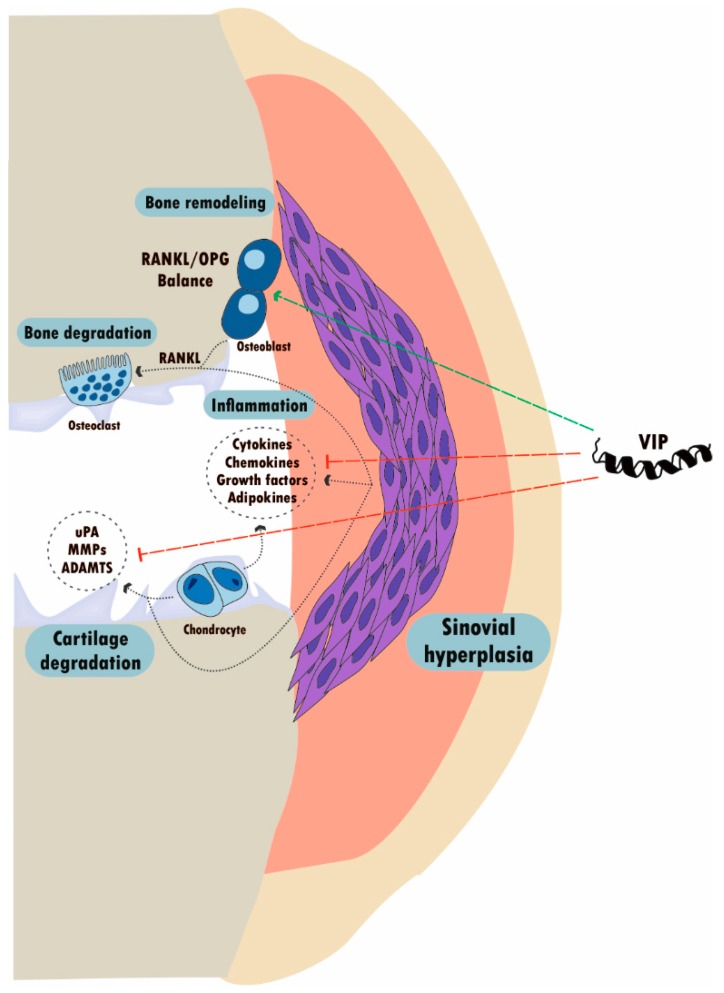
VIP effects in OA. Schematic overview of VIP effects in OA. Green arrows indicate “induction” whereas red arrows indicate “inhibition”.

**Figure 4 ijms-21-00065-f004:**
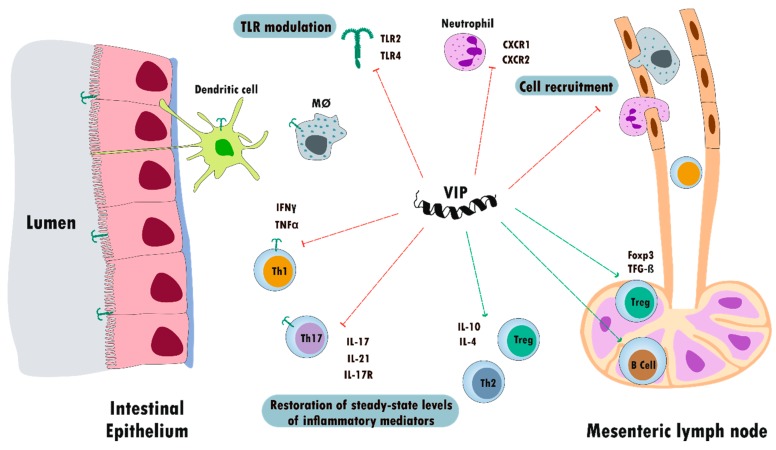
Biological effects of VIP in the TNBS-induced murine model of Crohn’s disease. Main effects of VIP on disease’s development are represented schematically. Green arrows indicate “induction”, whereas red arrows indicate “inhibition”.

**Figure 5 ijms-21-00065-f005:**
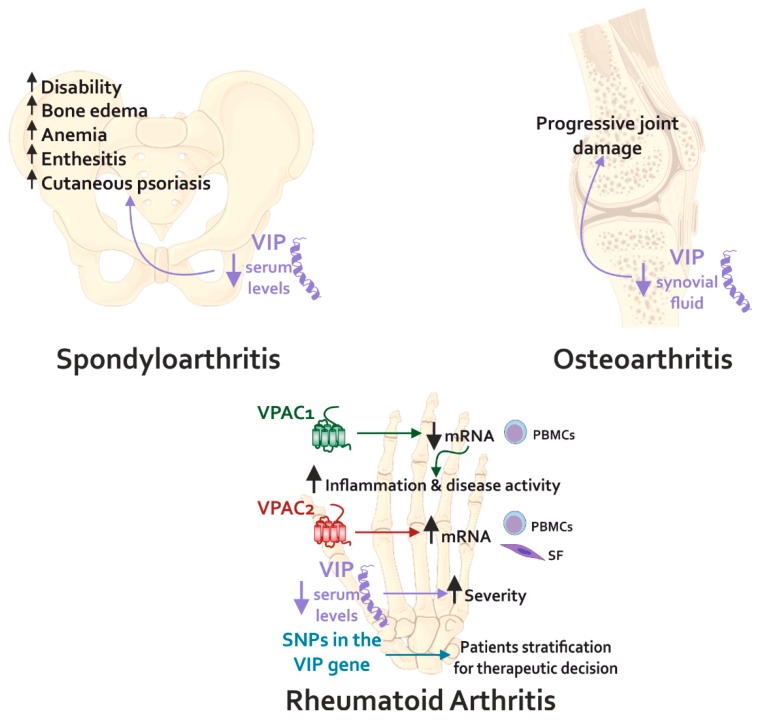
VIP and its receptors as biomarkers in rheumatic diseases. The scheme shows the current advances on the role of the VIP axis and VPAC receptors as biomarkers in spondyloarthritis (SpA), osteoarthritis (OA), and rheumatoid arthritis (RA). ↑Higher. ↓Lower. Purple arrows: association of VIP levels; green arrow: association of VPAC1 expression; red arrow: VPAC2 expression; blue arrow: clinical utility of SNPs (single nucleotide polymorphisms) in VIP gene.

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
