# Peer review of "A Clinical Approach for the Use of VIP Axis in Inflammatory and Autoimmune Diseases"

_ijms, 2019, doi:10.3390/ijms21010065_

Round 1

Reviewer 1 Report

This manuscript gives a comprehensive overview of Vasoactive intestinal peptide (VIP) that is involved on innate and adaptive immune responses. This is really outstanding review, that focuses on the biology of VIP and VIP receptor signaling, as well as its protective effects as an immunomodulatory factor.

The manuscript is well written, well structured and contains a wealthy amount of information on the VIP and its physiological roles. I did not find any weaknesses. Just two minor remarks:

Page 12 line 483 The additional word “detailed” should be deleted

(…) effects of VIP in the CIA model detailed above detailed

Page 12 line 497 The abbreviated form of matrix-metalloproteinase-1 should be (MMP-1) not (MMP)-1

Author Response

Dear Reviewer,

Thank you for revising our manuscript entitled “A clinical approach for the use of VIP axis in inflammatory and autoimmune diseases”. We have addressed all points raised by you as follows:

We have corrected the mistakes found by the referee in the corresponding sections.

We thank the opportunity to defend our article, and we hope that the revisions improve our manuscript.

Yours sincerely,

Carmen Martínez

Reviewer 2 Report

In this paper, Martinez et al. nicely highlight known and recent findings about the role of VIP/VPAC system in inflammatory and autoimmune diseases. The way it is written makes it accessible to a large scientific audience, who is not familiar with the VIP / VPAC system or the immune system. The paper summarizes well and discusses all data collected both through animal models (KO mice, disease models) but also from human samples.

Minor comments :

At lanes 293-294, the authors wrote that the importance of VIP in immune homeostasis was confirmed in KO mouse models. I would add here a sentence specifying that at basal conditions the immune phenotype of the mice studied so far is relatively mild. The role of VIP is mainly highlighted in challenging/inflammatory conditions.

There are some typo errors throughout the manuscript for example in lanes : 14,101, 145, 484, 602, 704, 705, 713, 873…

Author Response

Dear Reviewer,

Thank you for revising our manuscript entitled “A clinical approach for the use of VIP axis in inflammatory and autoimmune diseases”. We have addressed all points raised by you as follows:

We have included the sentences suggested by the referee (lanes 298-300). We have revised the manuscript to correct typographical errors.

We thank the opportunity to defend our article, and we hope that the revisions improve our manuscript.

Yours sincerely,

Carmen Martínez

Reviewer 3 Report

The article is interesting, well documented and well organized. Nevertheless, it is unclear why the authors chose to describe the role of VIP in OA over SLE/systemic sclerosis/inflammatory myopathies/SpA (seeing as the title orients towards inflammatory and autoimmune diseases). 

Author Response

Dear Reviewer,

Thank you for revising our manuscript entitled “A clinical approach for the use of VIP axis in inflammatory and autoimmune diseases”. We have addressed all points raised by you as follows:

According to the referee, we have included two phrases that may clarify the reason why we chose OA over other inflammatory and autoimmune rheumatic disorders (lanes 553-555).

The main reasons that have led us to treat OA are:

- It is the second most studied rheumatic disease in terms of the role of the VIP.

For example, in spondyloarthritis, there are 7 articles that appear in a search in Pubmed. Two of them correlate VIP levels and clinical parameters in SpA (one of our research group mentioned in the section Potential of VIP axis as a biomarker for personalized treatment in rheumatic diseases) and five in relation to a sequence from VPAC1 and its association to HLA-B27. Similarly, there are few data on systemic lupus erythematosus. Being the most prominent articles: one in which VIP reduces proteinuria and renal function defects and restores the Th17/Treg cell balance in a pristane-induced lupus mouse model and two in which the presence of autoantibodies against VIP in mouse models and in patients is described.

- It is currently also considered as a low-grade inflammatory disease and it is the most prevalent rheumatic disease, as we comment in the text.

We thank the opportunity to defend our article, and we hope that the revisions improve our manuscript.

Yours sincerely,

Carmen Martínez